# DNA Methylation and Telomeres—Their Impact on the Occurrence of Atrial Fibrillation during Cardiac Aging

**DOI:** 10.3390/ijms242115699

**Published:** 2023-10-28

**Authors:** Arkadiusz Grzeczka, Szymon Graczyk, Pawel Kordowitzki

**Affiliations:** Department for Basic and Preclinical Sciences, Faculty of Biological and Veterinary Sciences, Nicolaus Copernicus University, Szosa Bydgoska 13, 87-100 Torun, Poland

**Keywords:** atrial fibrillation, telomeres, telomere length, mDNA, methylation, age, aging, chronological age, epigenetic age

## Abstract

Atrial fibrillation (AF) is the most common arrhythmia in humans. AF is characterized by irregular and increased atrial muscle activation. This high-frequency activation obliterates the synchronous work of the atria and ventricles, reducing myocardial performance, which can lead to severe heart failure or stroke. The risk of developing atrial fibrillation depends largely on the patient’s history. Cardiovascular diseases are considered aging-related pathologies; therefore, deciphering the role of telomeres and DNA methylation (mDNA), two hallmarks of aging, is likely to contribute to a better understanding and prophylaxis of AF. In honor of Prof. Elizabeth Blackburn’s 75th birthday, we dedicate this review to the discovery of telomeres and her contribution to research on aging.

## 1. Introduction

Atrial fibrillation (AF) is the most common arrhythmia in humans [1,2,3]. Nations differ in their prevalence of AF as follows: 1.7% in Australia and New Zealand [4], 1.1% in the USA [5], 1.4% in the UK [6], and 1.6–1.7% in China [7]. AF is characterized by irregular and increased atrial muscle activation. This rapid muscle activation obliterates the synchronous work of the atria and ventricles, reducing myocardial performance, which can lead to severe heart failure or stroke [8]. The risk of developing atrial fibrillation depends largely on the individual patient’s history [9]. Therefore, AF is a consequence of the presence or accumulation of risk factors such as unhealthy body composition, physical inactivity or excess physical activity, old age, coronary artery disease, diabetes, hypertension, or valvular heart disease [2,9,10,11,12,13,14,15]. In addition, it can be caused by gene mutations that result in a very early onset of atrial fibrillation [16,17,18,19]. The pathophysiological basis for the development of atrial fibrillation is still not fully clear. Several theories include ion-flow disruption, structural remodeling, and electrical remodeling [9,20,21,22]. The formation of ectopic foci of excitation and, thus, the knotting of reentry loops, the intensity of proliferation, differentiation, and the remodeling of the extracellular matrix, as well as the efficiency of flow through ion channels, are associated with changes in gene expression [23,24,25,26]. Some changes in gene expression related to signaling pathways, fibroblast activity, and electrical conductivity, or associated with the development of inflammation in AF patients, have been identified [27,28,29,30,31,32,33]. However, it should be emphasized that the reading of genetic information is tightly controlled at each stage and can be up- or down-regulated. Modifications of expression via epigenetic processes are considered particularly important [34,35,36,37,38]. Cytosine methylation is the main mechanism underlying the epigenetic regulation of target genes. The methylation of cytosine occurs in the so-called CpG islands or their shore regions, which are areas encompassing 0–2 kbp. Through cytosine methylation, the availability of a given gene for expression is altered [39]. This process is catalyzed by methyltransferases (DNMTs). DNMT activity has been demonstrated in both DNA and RNA molecules. mDNA is responsible for stimulating transcription or silencing it. RNA modifications correspond to the level of post-transcriptional control of gene expression. It also appears that atrial tissue has a specific level of methylation and is more hypomethylated compared to ventricular tissue [40,41]. It has been shown that the development of diseases reorganizes the density of the methylome in the atria [40]. Thus, information about the characteristics of the methylome can be marked as clinically relevant.

Atrial fibrillation significantly contributes to reduced quality of life and carries many adverse consequences. Its epidemic extent makes AF one of the greatest threats to cardiovascular health today and has significant societal impacts. In less than 50 years, the likelihood of developing atrial fibrillation has nearly quadrupled [42]. This may be related to the development of diseases in civilization, but it is especially due to the higher average age of developing and developed countries [43]. Patients with higher chronological age (CA) have a higher incidence of AF, so it is thought that an aging population could contribute to an increased incidence of AF in the future [8,44]. The threshold CA value, after which the likelihood of AF increases significantly, is considered to be >65 years in Europe and America. A 5-year increase in CA is thought to be associated with a 1.44-fold higher risk of AF [45]. Related to this is the fact that, at this age, patients are simultaneously at greater risk for a number of diseases that can predispose them to atrial fibrillation [7]. However, pre-existing diseases, lifestyle (diet and the intensity of physical activity), occupational stress, life satisfaction, exposure to environmental stressors, and many other factors can affect another important parameter of AF exposure: biological aging. This effect, i.e., the discrepancy between chronological age and biological age, is called epigenetic age acceleration (EAA) [46]. EAA can be positive or negative, making biological age higher or lower than CA. Biological age is measured by epigenetic clocks, which measure DNA methylation and may also include, for the increased efficiency of measurements, phenotypic health parameters. Telomere attrition is one of the so-called hallmarks of aging and consists of repetitive G-rich sequences (TTAGGG) (Figure 1). Telomere attrition and dysfunction have become a well-established pathway involved in organismal aging and age-related diseases, not only because it imposes a limitation on cell division and tissue regeneration but also because telomere homeostasis influences other pathways involved in aging. In this review, we would like to show how, in the current state of research, the incidence, type, and progression of AF are related to age and the processes controlling it.

## 2. The Role of Epigenetics in the Development of AF

Underlying these methylome changes is the ability of methyltransferase enzymes (DNMT1, DNMT2, DNMT3A, and DNMT3B) to catalyze the reaction that transfers the methyl group to cytosine or adenosine. A group of these enzymes can regulate gene expression both at the transcriptional level and afterward [39,47]. Changes in gene methylation observed at different stages of AF development have been indicated as having a significant effect on the development of risk factors. In AF, decreasing the efficiency of atrioventricular flow contributes to an increase in left atrial (LA) pressure. An increased LA diameter is the result of increased atrial pressure and correlates with increased global mDNA [48]. The stretching of the atrial wall triggers atrial natriuretic peptide (ANP) activity and LA remodeling. The promoter site of the natriuretic peptide A gene (NPPA) contains nine CpGs that can be methylated [49]. ANP, therefore, could become a good marker for AF if the exact way in which ANP is regulated is understood [50]. It turns out that there is a positive correlation between the level of the natriuretic peptide A receptor (NPRA) gene promoter, methylation, and DNMT3B in AF [51]. As a result of this correlation, NPRA mRNA was significantly reduced in the AF group [51]. Increasing blood levels of pro-ANP in AF states is due to the role of atrial peptides in inhibiting aldosterone synthesis and renin secretion [52]. ANP receptors play an important role in the development of atrial fibrillation, as they have cardioprotective as well as anti-fibrotic properties; therefore, DNMT activity against ANP-related genes can contribute to the rapidity of fibrosis [53]. Their relationship to other factors, such as angiotensin-converting enzyme 2 (ACE2), that are involved in AF development is being studied [54]. ACE2 are mediators of inflammation and are involved in the electrical and structural remodeling of the heart [55]. ACE2 promoted the hypermethylation of individual genes [56] and, in mice, the concomitant administration of an ACE2 inhibitor and an Natriuretic Peptide Receptor-C (NPR-C) agonist reduced fibrosis [54]. ACE2 is an element of the renin-angiotensin system (RAS), whose role in the development of fibrosis has already been demonstrated [57,58]. Fibroblast activity can be inhibited by cell-cycle arrest due to the inhibition of cyclin D accumulation [59,60]. RAS-association domain family member 1 (RASSF1A), which inhibits cyclin D1 accumulation in activated fibroblasts but undergoes hypermethylation through DNMT3A activity in AF, is being silenced [33,61]. In human-induced pluripotent stem cells where DNMT3A knockout was performed, the cultured tissue exhibited impaired energy metabolism and degenerative changes in the mitochondrial body [41].

Patients with AF also exhibit impaired mitochondrial function [62], which can lead to the development of myocardial fibrosis [63]. Therefore, DNMTs are indispensable not only in pathophysiological processes but are also essential for proper cell functioning. Fibrosis is one of the components of myocardial remodeling. This occurs mainly via increasing fibroblast activity, but cardiac myoblasts and extracellular matrix modulators (MMPs and TIMPs) are also involved [64]. This process may be mediated by DNMT1, whose inhibition blocks the proliferation of fibroblasts [33]. By contrast, potent DNMT1 can drive fibroblast activation via the cardiac suppressor of the cytokine signaling-3 (SOCS3) axis and through the effects of IL6, IL10, and interferon (IFN)-gamma [65,66]. In a model of myocardial infarction, the application of SOCS3 conditional knockout (SOCS3-CKO) in mice significantly reduced the size of myocardial remodeling [67]. SOCS3 is also responsible for wound healing, and it has been indicated that miR-200b contributes to this process [68]. This may suggest that a certain program of methyltransferase activity regulates several pathways responsible for myocardial repair; in this context, studies in rats have indicated that DNMT3A acts by inhibiting miR-200b, which, in turn, inhibits fibroblast autophagy [69]. DNMT3-mediated regulation also involves ERK1/2, a well-known factor that stimulates cell proliferation and differentiation [33]. In AF, we observed that integrin and transforming growth factor (TGF)-β were hypomethylated, which could suggest their association with cardiac fibrosis [48,70,71].

The activity of DNMTs reveals differences in global DNA methylation in AF patients compared to the controls [51]. At the same time, increased methylation correlates with chronological age gain. However, sometimes, methylation does not maintain a strong association with CA while maintaining an association with AF [72]. This is why epigenetic changes seem to be such a promising marker. In genome-wide association studies (GWAS), changes in methylation levels has been shown in hundreds or thousands of CpGs [48,72]. In one study, 412 hypermethyl- and 450 hypomethyl-modified CpGs were detected, and most were located in regions with encoding genes [48]. Of this large number of altered CpGs, ultimately, few remain associated with AF. In the case of post-operative AF (PoAF) analysis, twelve PoAF-associated CpGs were detected, and in another study for the same type of AF, researchers found two CpGs [72,73]. The relationship between these two CpGs for persistent atrial fibrillation (PsAF) and the five identified for incident AF was also determined [74]. As listed by Donate et al., changes in gene expression affect proteins involved in protein localization and catabolism, morphogenesis and development, signaling, metabolism, and cell death [75]. Thus, a very large number of genes is subject to altered expression in AF, mainly upstream [75]. Statistically significant changes in methylation involving the silencing of gene expression suggest an association between DNA hypermethylation and AF. Expression silencing includes paired-like homeodomain 2 (PITX2), in which one of its roles is to promote atrial muscular development within the atrial entry for pulmonary veins: a susceptible site for AF initiation [76]. Over the years, studies have indicated the increasingly important role of PITX2 in the development of heart disease, including AF [17,77]. GWAS indicated that there may be one or two single nucleotide polymorphisms (SNPs) on chromosome 4q25 near the PITX2 gene that are associated with a higher risk of AF: rs10033464 and rs2200733 [78]. It was also indicated that 35% of Europeans have at least one of these [78]. The relationship between the increased methylation of the PITX2 promoter, increased DNMT1, and reduced PITX2c protein levels have also been demonstrated [56]. In addition, human atrial myocytes cultured from knockout stem cells (DNMT3A gene knockout) led to the increased expression of PITX2 [41]. A short time later, hypermethylation was indicated in both the CpG and shore region at the PITX2c locus [48]. However, of these two islands whose methylations were detected, only one (CpG24) was identified as a possible regulator of PITX2 expression [75]. The hypermethylation of the PITX2 promoter site is also influenced by increased levels of ACE2, thereby helping drive the fibrosis of atrial tissue [33].

Generally, the changes in methylation of individual SNPs associated with AF, of which there are more than 100 [25,78,79,80,81,82,83], are poorly understood. Previously identified associations between twelve SNPs and AF were analyzed for their association with CpG methylation [74]. This study indicates that each SNP is significantly associated with at least one methylation [74]. The most potent association was shown by the cg10833066 (CpG site) with the rs6490029 loci for CUX2. The gene located here encodes a transcription factor that is responsible for cell proliferation and differentiation and can undergo differential methylation, as can other transcription factors [40]. In the nervous system, it controls the proliferation of neuronal precursors, but its role in cardiac tissue, let alone in the development of AF, is not clear [84,85]. For example, cg13639451, with a significant change in methylation, is located close to the inhibitor of nerve–fiber development. In turn, cg07191189 is located even closer, and another striatin gene, STRN, that is part of the SLMAP/Striatin complex, is in part responsible for electrical conduction [74,86]. There are also significant decreases in the methylation of islands close to genes whose function is unknown [72]. Another transcription factor, RUNX family transcription factor 1 (RUNX1), has been shown to interact with other differentially methylated genes [48]. Other CpG gene associations include cg13639451—associated with the expression of ribosomal protein S18 (RPS18)—which is involved in translation initiation, and cg15440392—associated with the expression of granzyme H (GZMH)—which is responsible for inducing the cell death of target cells [74]. The multitude of genetic variations that are identified with a predisposition to AF has contributed to the creation of the polygenic risk score (PRS), which can be used to determine exposure on the basis of cumulative genetic effects [87].

## 3. Epigenetic Clocks and AF

The biological age that epigenetic clocks determine is based on the value of genome methylation profiles. Changes in methylation directly affect the final result that the epigenetic age (EA) calculation achieves. Epigenetic clocks can be categorized based on the tissue they use to determine age. The Hannum blood cell DNA clock is based on 71 CpGs, while Horvath’s clock, which is based on 353 individual CpG probes, allows the use of various tissues [88]. However, due to the different methylome of each tissue, Horvath’s clock requires validation each time a new tissue is measured [89]. Importantly, Horvath’s epigenetic clock is almost zero in embryonic cells and pluripotent stem cells. By contrast, the association of methylation with cell development and division causes the epigenetic clock to have a high rate at a low CA. Then, after the organism reaches adulthood, it achieves a constant rate. By contrast, according to Horvath et al. (2018), the Hannum clock is better for determining the effects of aging, including mortality [46]. Newer clocks, such as DNAm PhenoAge, not only measure biological age (Figure 2) on the basis of the methylation of CpGs (513) but also link the analysis to certain parameters such as albumin, creatinine, glucose (serum), C-reactive protein, lymphocyte percent, mean cell percent, red cell distribution width, alkaline phosphatase, and white blood cell count [90]. By contrast, the clock that includes the most CpGs is DNAm GrimAge. This determines biological age based on 1030 CpGs as well as the following: adrenomedullin, C-reactive protein, plasminogen activation inhibitor 1 (PAI-1), and growth differentiation factor 15 (GDF15) [91]. To make the GrimAge estimator as precise as possible, a DNAm packet-smoking estimator was also added due to the significant risk of cigarette smoking [91]. Clocks have previously been linked to the incidence of neurological diseases, genetic diseases, and, more recently, cardiovascular diseases, including atrial fibrillation [45,92,93,94,95,96] (Figure 2).

Still, few studies have been devoted to the link between AF and the ticking epigenetic clock despite the fact that heart disease can significantly accelerate the epigenetic clock [94]. The only available studies are based on five epigenetic measures [45]. It was indicated that a five-year increase in CA resulted in a 1.26 to 1.43 increase in the probability of AF, depending on the test. Accelerated epigenetic aging was associated with AF, most strongly when it was calculated using GrimAge, and AF was unrelated to Horvath clock estimates. After adjusting for risk factors, significance persisted only for GrimAge and PhenoAge. However, Mendelian randomization, when performed in the same study, did not reveal causal relationships. Heart failure also showed no association with EA [94]. By contrast, patients with ischemic stroke, a common consequence of AF, were epigenetically older by 4.6 years (Horvath clock) or 3.3 years (Hannum clock) compared to their CA age [95].

## 4. Telomeres and AF

Environmental factors, oxidative stress, and inflammation contribute to telomere shortening, the length of which is a different method of assessing AF risk and determining age. In fact, telomeres are repeats of the sequence TTAGGG. This structure localizes at the ends of DNA strands and is a way to protect valuable genetic material from damage during the compaction and relaxation of chromatin during the numerous complex cycles of cell division [97]. The length of telomere shortens with each division until it becomes too short to allow the cell to divide safely [62]. This is when programmed cell death occurs [98,99]. However, as telomeres shorten (both as a result of programmed aging but also as a result of acceleration due to external factors), p53 expression is activated, thereby inhibiting peroxisome proliferator-activated receptor γ coactivator-1 (PGC-1) and causing mitochondrial dysfunction [62]. Energy deprivation and the accumulation of reactive oxygen species can induce AF. Measurements on the telomere length of blood leukocytes are most commonly used since a relationship has been shown between peripheral blood telomere length and telomere length in other tissues [62]. Myocardial tissue studies are also being conducted to accurately determine the relationship between AF and the relative telomere length (rTL).

In most studies, telomere length is inversely correlated with chronological age [62,100,101]. This was demonstrated in both study groups: those with AF and the healthy control group [62]. However, some have not shown such a relationship [102]. Another contradiction concerns the relationship between rTL and AF. In addition to studies showing a negative association between rTL and atrial fibrillation [103], there are studies that show a positive association between rTL and atrial fibrillation, as well as those showing no association between rTL and atrial fibrillation [104,105,106]. In addition, incomplete knowledge of the relationship between CA and rTL and the development of AF is supported by studies that show an association between AF and rTL in younger groups with AF but no association in older groups with AF [107]. As Pan et al. correctly noted, rTL shortening, especially excessive shortening, may play an important role in the development of atrial fibrillation in young (<50) individuals. This relates to younger people’s greater exposure to occupational hazards, stress, or higher alcohol consumption [108,109,110]. In addition, when analyzing studies of cardiac electrical activity, there is no correlation between electrocardiographic features on the ECG and rTL [111]. However, those with a history of AF were significantly older than those free of arrhythmia and had shorter telomeres [62,111,112,113]. Therefore, to show a trend between the association of AF and rTL, age stratification is used (the entire study group was divided into quartiles—Q1–Q4) [112]. The rTL of patients with AF was mainly located in Q1 and Q2, i.e., at a lower rTL, while control subjects were in Q3 and Q4 [62,112]. Statistical results are often influenced by many factors, and there is a risk of a high rate of latent AF. Although several works have succeeded in determining the relationship between AF and rTL, rTL can not always be said to be an independent factor [100,112,113,114,115]. Cox’s multivariate proportional hazards model allows for the inclusion of concomitant events that may interfere with the true picture of this relationship. In the case of AF, factors such as age, gender, body mass index, hypertension, or diabetes are considered. As a result of taking risk factors into account, in some cases, rTL maintained or lost its association with AF, or there was a reduction in significance [101,112]. The same test was used to assess the predictive power of rTL [62]. It has been shown that rTL can be an independent predictor of the onset of AF, the progression of paroxysmal AF (PAF) to persistent AF, and a predictor of recurrent AF (RAF) [103,113,115]. The few papers on RAF unanimously show that patients with recurrent atrial fibrillation have a significantly shorter rTL than patients without RAF [105,106,107]. Also, people with a shorter rTL are more likely to develop RAF [105]. However, it has not always been shown to be independent when using multivariate Cox logistic regression [105,107]. However, it was possible to determine in groups with and without RAF a cutoff value equal to t/s = 1.040 and how exceeding this value predisposed individuals to RAF [105]. RAF is one of the factors that can predispose individuals to arrhythmia fixation. Distinguishing AF types based on rTL has not been widely studied. According to some, there are no significant differences in rTL between AF types [62]. However, rTL can reach differential values and be associated with a specific type. One study distinguished persistent atrial fibrillation (PsAF) from AF in groups where CA in both was comparable [103]. The cut-off point was calculated to be 1.175 with a sensitivity of 56.03 and a specificity of 82.04% [103]. The authors indicated that patients who progressed with AF were characterized by a higher rTL [103]. The difference between PAF and PsAF was also confirmed by others [113]. There was no significant difference in rTL between the group without AF and the PAF group [115]. Also, in the case of permanent AF, no significant differences were indicated between it and PsAF in rTL [112]. At the same time, rs2736100, an SNP site lying near the TERT gene (responsible for the expression of the telomerase reverse transcriptase enzyme), has not been shown to be associated with the occurrence of AF [104]. According to the authors, this indicates no association between rTL and AF risk. Recently, several authors have presented meta-analyses of data on the association between rTL and AF, as well as publications analyzing the cause-and-effect relationship in randomized Mendelian trials. These may partially explain the contradictions present in the association between rTL and AF. rTL, according to Zheng et al. 2022, as a continuous univariate and multivariate variable, appears to be associated with AF risk; however, this relates to RAF and not new cases of AF [116]. In addition, rTL has been shown to be related to the progression of AF to PsAF. In summary, their study indicates the relevance of rTL for secondary forms of AF (RAF, PsAF). This may suggest that accelerated telomere shortening is secondary to the onset of AF. This year, a Mendelian analysis was presented, indicating such a cause-and-effect relationship [117]. The consequences of AF incidence, however, make this novel inverse relationship difficult to explain. Atrioventricular dyssynchrony, which occurs during AF incidents, results in the reduced efficiency of blood flow, thereby increasing the occurrence of failure syndromes of varying importance to the life and homeostasis of the body and is microscopically associated with mitochondrial dysfunctions. These disorders can cause telomere shortening. It is noteworthy that despite the relationship between the telomeres length of atrial cells (ATL) and rTL in peripheral blood, atrial cells divide much less frequently, and it is also a fact that ATL is, on average, larger than rTL [104]. The above information supports the conclusion that telomeres are directly related to cell division and may not necessarily always be an AF trigger. In addition, there is the possibility that telomere shortening is more important at younger ages as an AF trigger due to the different etiologies in young and old age [118].

## 5. Conclusions

Current studies do not provide a complete picture of how and to what extent AF is related to DNA methylation, epigenetic age, and telomere length. The results and numerous clinical implications indicate that these are important processes, but several obstacles stand in the way of understanding them. For both methods of estimating biological age, it is necessary to accept some common constraints, which limit research success as follows: the methods for measuring and recording AF (the presence of latent AF), the duration of observation, and the population studied.

Nonetheless, existing research allows for several conclusions and perspectives to be drawn. The hypermethylation of individual genes in atrial cardiomyocyte cells is also accompanied by global hypermethylation in the body, so further genome-wide DNA methylation profiling studies in conjunction with in situ studies are needed. 

As indicated, significant DNA methylation changes are present in key SNPs and are correlated with AF risk. Despite few data linking mDNA to a specific type of AF, it has been shown that the mDNA profile significantly differs between the pre-operative and post-operative periods and is associated with the occurrence of PoAF. However, studies on specific populations are not representative of the entire population. The promising value of mDNA age is its close correlation with CA, and the active involvement of DNMTs in signaling pathways belonging to AF pathophysiology allows us to conclude that this topic should become one of the most important for achieving not only cognitive but also therapeutic goals in the context of AF. 

In contrast to epigenetic clocks, telomere length correlates with chronological age to a limited extent, as confirmed by various studies. In addition, telomeres are also subject to other processes besides shortening, which may affect the interpretation of the results [119]. However, rTL has a predictive value for secondary forms of AF. Recurrent AF or the perpetuation of AF are forms whose occurrence could be predicted by telomere shortening. Therefore, this indicates its value in patients already hospitalized.

## Figures and Tables

**Figure 1 ijms-24-15699-f001:**
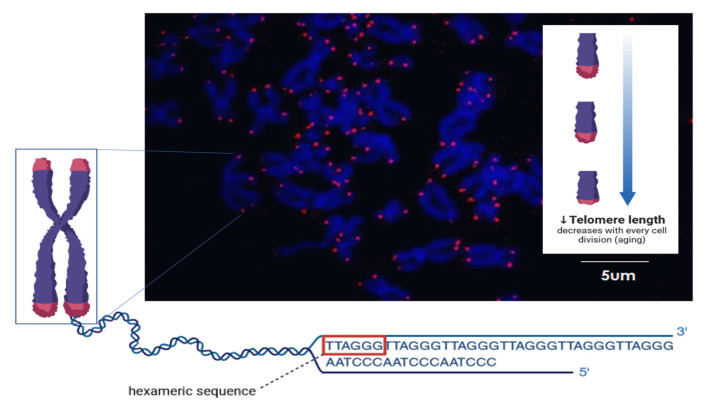
Schematic showing the hexameric sequence (TTAGGG) of telomeres and a confocal picture in which a spread of chromosomes (blue) in metaphase is shown with stained telomeres (red). The arrow indicates the decrease of telomere length. The figure has been created with biorender.com (accessed on 10 June 2023).

**Figure 2 ijms-24-15699-f002:**
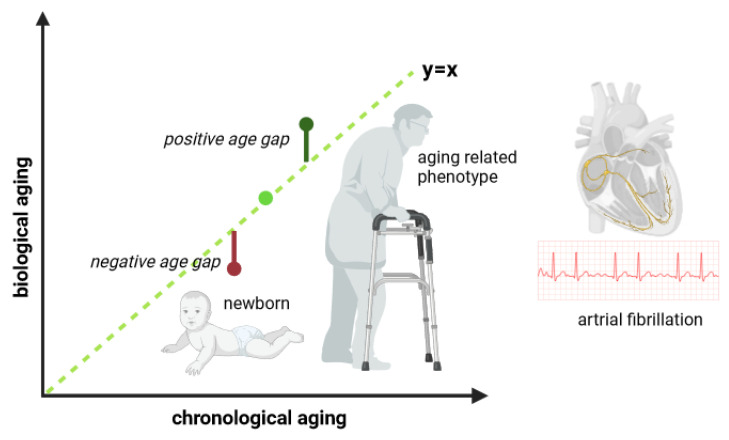
Schematic showing the potential differences between chronological aging (*x*-axis) and biological aging (*y*-axis). Biological age could vary in relation to aging-related phenotypes and/or diseases; for instance, due to a healthy lifestyle a person could have a younger biological age compared to their chronological age (positive age gap), or, due to an unhealthy lifestyle and disease, a person could biologically age more, and therefore, have a higher biological age compared to chronological age. The figure has been created with biorender.com (accessed on 10 June 2023).

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
