# Peer review of "DNA Methylation and Telomeres—Their Impact on the Occurrence of Atrial Fibrillation during Cardiac Aging"

_ijms, 2023, doi:10.3390/ijms242115699_

Round 1

Reviewer 1 Report

Comments and Suggestions for Authors

The authors provide an overview on the literature concerning the role of telomere length and DNA-methylation in the prediction and occurrence of atrial fibrillation, a subject of high scientific interest.

I have the following concerns:

1)      The manuscript is lacking a figure synthesizing the findings.

2)      Most abbreviations are not defined in the text. I had to check the references to find their meaning: mDNA, PoAF, PsAF, LTL, TL, PAF.

3)      Line 45: what does “lying along them” mean?

4)      Lines 70-71: this sentence has to be changed to make it understandable

5)      In general, the use of  commas should be checked (for example: line 107 “inhibited by, cell cycle” makes no sense)

6)      Line 102: “ACE” is not the abbreviation of angiotensin (ACE = angiotensin converting enzyme)

7)      Lines 108-109: “, which” should be deleted;

8)      Lines 146-148: this sentence is not understandable

9)      Line 160: I do not understand where CpG24 comes from?

10)   Epigenetic clocks are not defined/described at the beginning of section 3. The fact that they are based on CpG methylation has to be guessed by the reader.

11)   Figure 2: the meaning of positive or negative age gaps should be described.

12)   Lines 244-247: this sentence is not understandable

13)   In line 258 the authors mention Q1-Q4 without explaining what this means. It is not up to the reader to read the references to understand the manuscript

14)   Lines 279-282: I have the impression that AF is the same as PAF? Is this true? If yes, than why use different abbreviations?

15)   Line 284: what is “fixed AF”?

Comments on the Quality of English Language

Problem with the English language is mostly a confusing structure of sentences and inappropriate use of commas

Author Response

MS ID: ijms-2648573

Dear Academic-Editors and Reviewers,

Thank you for inviting us to respond to the very thoughtful and constructive reviewer comments. We greatly appreciate the reviewer's time and believe our revised manuscript has become more well-rounded as a result.

We have incorporated all suggestions throughout the manuscript, and the changes are highlighted in "track changes”. Below is a point-by-point response to the reviewers’ and Editor’s comments to clarify which edits were made.

We are happy to respond to additional requests if they arise.

Sincerely,

Dr. Paweł Kordowitzki

Please note our following explanations:

Detailed answers to Reviewer 1

REVIEWER: The manuscript is lacking a figure synthesizing the findings.

ANSWER: We thank the reviewer for this suggestion. In response, we have decided to add a graphical abstract.

REVIEWER: Most abbreviations are not defined in the text. I had to check the references to find their meaning: mDNA, PoAF, PsAF, LTL, TL, PAF.

ANSWER: Abbreviations have been standardized and explained.

REVIEWER: Line 45: what does “lying along them” mean?

ANSWER: The awkward wording has been removed. Line 44-45.

REVIEWER: Lines 70-71: this sentence has to be changed to make it understandable

ANSWER: The sentence was corrected and clarified: “EAA can be positive or negative, making biological age higher or lower than CA.” Line 83-84.

REVIEWER: In general, the use of commas should be checked (for example: line 107 “inhibited by, cell cycle” makes no sense)

ANSWER: Unnecessary commas have been reduced and this has been indicated by "tracing changes."

REVIEWER: Line 102: “ACE” is not the abbreviation of angiotensin (ACE = angiotensin converting enzyme)

ANSWER: The incorrect abbreviation has been removed. Line: 124.

REVIEWER: Lines 108-109: “, which” should be deleted;

ANSWER: The word "which" has been removed.

REVIEWER: Lines 146-148: this sentence is not understandable

ANSWER: The sentence was corrected and clarified: Statistically significant changes in methylation involving silencing of gene expression suggest an association of DNA hypermethylation with AF.” Line: 184-185

REVIEWER: Line 160: I do not understand where CpG24 comes from?

ANSWER: The sentence was corrected and clarified: However, of the two islands whose methylations were detected, only one (CpG24) was identified as one of the possible regulators of PITX2 expression.” Line 198-199.

REVIEWER: Epigenetic clocks are not defined/described at the beginning of section 3. The fact that they are based on CpG methylation has to be guessed by the reader.

ANSWER: We have added a passage explaining the relationship between epigentic age and DNA methylation: “The epigentic ages that epigenetic clocks determine are based on the value of ge-nome methylation profiles. Changes in methylation, directly affect the final result that the EA calculation achieves.” Line: 238-240.

REVIEWER: Figure 2: the meaning of positive or negative age gaps should be described.

ANSWER: We thank the Reviewer for this comment. In response we have improved the figure legend as follows:

Scheme showing the potential differences between chronological aging (x-axis) and biological aging (y-axis). Biological age could be different in relation to aging-related phenotypes and/or diseases, for instance, due to a healthy lifestyle a person could have a younger biological age compared to the chronological age  (positive or negative age gap), or due to an unhealthy lifestyle and diseases a person could biologically age more, and therefore, has a higher biological age compared to chronological age.

REVIEWER: Lines 244-247: this sentence is not understandable

ANSWER: The sentence was corrected and clarified: “In addition to studies showing a negative association between rTL and atrial fibrillation [104], there are studies showing a positive association between rTL and atrial fibrillation, as well as those showing no association between rTL and atrial fibrillation [105–107].” Line 303-306

REVIEWER: In line 258 the authors mention Q1-Q4 without explaining what this means. It is not up to the reader to read the references to understand the manuscript.

ANSWER: We have added a passage explaining: (The entire study group was divided into quartiles - Q1-Q4)

REVIEWER: Lines 279-282: I have the impression that AF is the same as PAF? Is this true? If yes, than why use different abbreviations?

ANSWER: In the passage where we write about PAF, the authors distinguished the presence of different types of AF. The works where we use AF, on the other hand, are mostly population-based studies, where it is difficult to precisely determine the procet of progression of one condition into another, hence also We, quoting them, cannot assume that it is only PAF or already PsAF.

REVIEWER: Line 284: what is “fixed AF”?

ANSWER: We changed to “permanent AF”.

Reviewer 2 Report

Comments and Suggestions for Authors

In fact, I found the manuscript conglomerate of asuperficial pieces of information. Although the authors claim that “In this review, we  would like to show how in the current state of research the incidence, type, and progression of AF is related to age and the processes controlling it.” There is no response to this declaration.

They also claim that “The mitotic clock, on the other hand, which can also reflect biological age, is a measure of cell division and relies on assessing telomere length (Fig. 1) and shortening rates.” I would not agree. How about single chromosome ends? The critical ones?

There are also some clumsy phrases like: “attaching a methylating molecule to the cytosine” What does it mean?

Fig 2 – difficult to follow – scant caption.

Regarding stress response – not only telomeres but also hTERT expression/localization matters.

1,6%-1,7% in Chinese [7] – should be in China.

cg10833066 island – it is not an island but just a site – that is a difference that the authors do not seem to understand all over the manuscript.

No explanation of LTL or mDNA abbreviations.

Comments on the Quality of English Language

Requires extensive editing of English language.

Author Response

MS ID: ijms-2648573

Dear Academic-Editors and Reviewers,

Thank you for inviting us to respond to the very thoughtful and constructive reviewer comments. We greatly appreciate the reviewer's time and believe our revised manuscript has become more well-rounded as a result.

We have incorporated all suggestions throughout the manuscript, and the changes are highlighted in "track changes”. Below is a point-by-point response to the reviewers’ and Editor’s comments to clarify which edits were made.

We are happy to respond to additional requests if they arise.

Sincerely,

Dr. Paweł Kordowitzki

Detailed answers to reviewer 2

REVIEWER: In fact, I found the manuscript conglomerate of asuperficial pieces of information. Although the authors claim that “In this review, we would like to show how in the current state of research the incidence, type, and progression of AF is related to age and the processes controlling it.” There is no response to this declaration.

ANSWER: We have rewritten the "conclusions" sections to respond to the assumptions made:

5. Conclusions

The current studies presented do not provide a complete picture of how and to what extent AF is related to DNA methylation, epigenetic age, and telomere length. The results and numerous clinical implications indicate that these are important processes, but several obstacles stand in the way of understanding them. For both methods of these estimators of biological age, it is necessary to accept some common limits that will limit research success, and these are: the methods for measuring and recording AF (the presence of latent AF), the duration of observation, and the population studied.

Nonetheless, existing research allows for several conclusions and perspectives. Hypermethylation of individual genes in atrial cardiomyocyte cells is also accompanied by global hypermethylation in the body, so further Genome-wide DNA methylation profiling studies in conjunction with in situ studies are needed.

As indicated, significant DNA methylation changes are present in key SNPs correlated with AF risk. Despite few data linking mDNA to a specific type of AF, it has been shown that the mDNA profile significantly differs between the preoperative and postoperative periods and is associated with the occurrence of PoAF. However, studies on specific populations will not be representative of the entire population. The promising value of mDNA ages is their close correlation with CA, and the active involvement of DNMTs in signaling pathways belonging to AF pathophysiology allows us to conclude that this topic should become one of the most important for achieving not only cognitive but also therapeutic goals in the context of AF.

In contrast to epigenetic clocks, telomere length correlates much weaker with chronological age, as confirmed by some studies. In addition, telomeres are also subject to other processes besides shortening, which may affect the interpretation of results [120]. However, rTL has predictive value for secondary forms of AF. Recurrent AF or perpetuation of AF were forms whose occurrence could be predicted by telomere shortening. Therefore, this indicates its value in patients already hospitalized.”

REVIEWER: They also claim that “The mitotic clock, on the other hand, which can also reflect biological age, is a measure of cell division and relies on assessing telomere length (Fig. 1) and shortening rates.” I would not agree. How about single chromosome ends? The critical ones?

ANSWER: We thank the reviewer for this criritcal comment. In response, we have decided to delete this sentence and replace it with the following text:

Telomere attrition is one of the so-called hallmarks of aging and consists of repetitive G-rich sequences (TTAGGG) (Fig. 1) Telomere attrition and dysfunction have become a well-established pathway involved in organismal aging, and aging-related diseases not only because it imposes a limitation on cell division and tissue regeneration but al-so because telomere homeostasis influences other pathways involved in aging.

REVIEWER: There are also some clumsy phrases like: “attaching a methylating molecule to the cytosine” What does it mean?

ANSWER: We changed our minds on:Through cytosine methylation”. Line: 58-59.

REVIEWER: Fig 2 – difficult to follow – scant caption.

ANSWER: We thank the Reviewer for this comment. In response we have improved the figure legend as follows:

Scheme showing the potential differences between chronological aging (x-axis) and biological aging (y-axis). Biological age could be different in relation to aging-related phenotypes and/or diseases, for instance, due to a healthy lifestyle a person could have a younger biological age compared to the chronological age  (positive or negative age gap), or due to an unhealthy lifestyle and diseases a person could biologically age more, and therefore, has a higher biological age compared to chronological age.

REVIEWER: Regarding stress response – not only telomeres but also hTERT expression/localization matters.

ANSWER: We have added a passage explaining:At the same time, rs2736100, a SNP site lying near the TERT gene (responsible for expression of the telomerase reverse transcriptase enzyme), has not been shown to be associated with the occurrence of AF [105]. Which, according to the authors, indicates no association of rTL with AF risk.” Line: 403-406.

REVIEWER: 1,6%-1,7% in Chinese [7] – should be in China.

ANSWER: It has been changed to the correct form. Line: 25.

REVIEWER: cg10833066 island – it is not an island but just a site – that is a difference that the authors do not seem to understand all over the manuscript.

ANSWER: Thank you for your attention. It has been corrected. Line: 210.

REVIEWER: No explanation of LTL or mDNA abbreviations.

ANSWER: Abbreviations have been standardized and explained.

REVIEWER: Requires extensive editing of English language.

ANSWER: Thank you for your attention. The "native" proofreader raised the quality of the language in the text.

Round 2

Reviewer 2 Report

Comments and Suggestions for Authors

Significantly improved.

Author Response

The minor comments have addressed.
